# Radiotherapy Plus the Neurokinin-1 Receptor Antagonist Aprepitant: A Potent Therapeutic Strategy for the Treatment of Diffuse Intrinsic Pontine Glioma

**DOI:** 10.3390/cancers17030520

**Published:** 2025-02-04

**Authors:** Miguel Muñoz, Marisa Rosso

**Affiliations:** Research Laboratory on Neuropeptides, Institute of Biomedicine of Seville (IBIS), 41013 Seville, Spain; marisarossog@gmail.com

**Keywords:** DIPG, glioma, substance P, neurokinin-1 receptor, aprepitant, antitumor, apoptosis, angiogenesis, metastases

## Abstract

Diffuse intrinsic pontine glioma (DIPG) is a devastating childhood brain tumor. The median survival of DIPG is 16 to 24 months, regardless of the treatment received. Therefore, new therapeutic strategies against DIPG are urgently needed. The peptide substance P (SP), through the neurokinin-1 receptor (NK-1R), is involved in glioma progression. Furthermore, all glioma cells express NK-1R, and NK-1R is essential for glioma cell viability. In contrast, NK-1R antagonists, such as the drug aprepitant, penetrate the brain and counteract all the pathophysiological effects produced by SP in glioma. The combination of radiotherapy with NK-1R antagonists produces radiosensitization and radioneuroprotection. This review updates the involvement of the SP/NK-1R system in glioma progression and the clinical application of NK-1R antagonist drugs in DIPG therapy. NK-1R plays a crucial role in glioma progression and NK-1R antagonists such as aprepitant could be used in combination with radiotherapy as a therapeutic strategy for DIPG patients.

## 1. Introduction

Glioblastoma or glioma is an astrocytoma of WHO grade IV. Diffuse intrinsic pontine glioma (DIPG) has been classified as a new pathologic entity called diffuse midline glioma, H3K27M-altered by the World Health Organization [1]. DIPG is a childhood malignancy of the brainstem with a dismal prognosis, and the median survival of DIPG is 16–24 months independent of the treatment received [2]. The combination of multiple cranial neuropathies and long tract and cerebellar signs is known as the “classic triad” of DIPG, and only a small percentage of pediatric patients (<10%) develop symptoms of hydrocephalus [2]. Recent studies analyzing the molecular profiles of DIPG found that most harbor a mutation in histone H3 (H3K27M-altered) and that they are more aggressive and have worse outcomes compared to their non-mutant counterparts, regardless of histology. Despite recent advances in the understanding of the molecular basis of the tumor in the last decade, the prognosis of DIPG has remained unchanged [2]. DIPG is resistant to conventional chemotherapeutic agents used for brain tumors, such as temozolomide. Radiotherapy (RT) is the standard treatment of DIPG; however, it is only palliative. RT is expected to increase survival for patients by about 3 months on average. DIPG remains universally fatal [3]. Unfortunately, new drugs only slightly improve overall survival in DIPG. Recently, a systematic review has been published about clinical trials on DIPG involving radiation therapy in combination with chemotherapeutic and/or included immunotherapeutic options, but even CAR T cell and oncolytic virus therapies showed a lacking response alongside both traditional and highly cytotoxic chemotherapeutic options [4]. Therefore, it is necessary and urgent to investigate a new and different molecular basis of DIPG tumor development and to search for new therapeutic targets, as well as new effective and safe drugs that can improve the ominous prognosis of DIPG.

In the last twenty years, considerable evidence has been published that substance P (SP) peptide and its receptor, neurokinin-1 receptor (NK-1R), are involved in glioma promotion and progression. Both NK-1Rs [5,6,7] and SP [7] are overexpressed in glioma tumors. SP, in a concentration-dependent manner after binding to NK-1R of glioma cells, increases tumor cell proliferation (mitogenesis), has antiapoptotic effects, and is involved in the Warburg effect. SP, after binding to NK-1R, increases (a) endothelial cell proliferation, (b) induces angiogenesis, (c) produces inflammation, and (d) induces migration and invasion (invasion and metastases) of glioma cells [8,9]. Moreover, NK-1R is essential for the viability of glioma cells and not of normal cells [10]. In contrast, NK-1R antagonists, in a concentration-dependent manner, counteract all the pathophysiological effects of SP in glioma: inhibiting glioma cell proliferation (antimitogenic effect), inducing programmed cell death in glioma cells (apoptosis), counteracting the Warburg effect in glioma cells, decreasing both endothelial cells’ proliferation and angiogenesis, providing anti-inflammatory effects, and preventing invasion and migration (infiltration and metastases) of glioma cells [8,9,11]. Furthermore, new neoadjuvant therapy is well established in neuro-oncology, and neoadjuvant local treatment of malignant gliomas using the peptidic vector [90Y]-DOTAGA–SP could be shown to be feasible [6]. However, due to the relatively low stability of SP and its fragments and analogues, new NK-1R ligands are constantly being investigated, including aprepitant, casopitant, and netupitant that have the potential to improve therapeutic monitoring and to explore the duration of pharmacological effects in the course of targeted radionuclide therapy [12]. Therefore, NK-1Rs could be considered a new therapeutic target in gliomas and NK-1R antagonists such as the drug aprepitant could be new selective, effective, and safe drugs in the treatment of gliomas such as DIPG.

## 2. SP and NK-1R in Glioma

SP and hemokinin-1 (HK-1) are members of the tachykinin peptide family, such as neurokinin A (NKA) and NKB. Tachykinins, via NK-1R, NK-2R, and NK-3R, exert many biological actions. Neurokinin A is the natural ligand of NK-2R and neurokinin B is the natural ligand for NK-3R. NK-1R, NK-2R, and NK-3R are three types of G protein-coupled transmembrane receptors. SP/HK-1 are known to have a preferential affinity for NK-1R, they are the natural ligands of NK-1R. The TAC1 gene encodes SP and TAC4 encodes HK-1 [13]. SP and HK-1 are two undecapeptides. The C-terminus sequence of SP 6–11 (Gln-Phe-Phe-Gly-Leu-Met) [14] is essential for affinity for the NK-1R. Since HK-1 has a C-terminus 6–11 similar to that of SP, it has a similar affinity for the NK-1R [15]. This suggests that NK-1R is very important for cells because it has two natural ligands, SP and HK-1. Glioma cells express both SP and HK-1 [16], however, SP is not essential for the viability of glioma cells [10], and the absence of SP is probably compensated by HK-1 in glioma cells. Furthermore, SP is ubiquitous throughout the body, in tissues, cells, and body fluids (blood, cerebrospinal fluid, saliva, etc.) [17]. However, SP is overexpressed in cancer tissues and cancer cells [18] and cancer patients have higher serum SP levels than normal subjects [19]. SP is located in cytoplasm and in the nucleus of glioma cells [10]. In addition, SP is well known to be a universal mitogen in cancer (including glioma) cells [9,11].

The NK-1R belongs to the class of seven transmembrane domain (7-TM) receptors and is functionally coupled with G protein-coupled receptors (GPCRs) and mediates the biological activities of SP or HK-1, which are the most potent agonists for the NK-1R compared to neurokinin A or neurokinin B [20]. This G complex consists of three different subunits: Gα induces an exchange of GTP for GDP and Gβ and Gγ subunits that form the complex Gβγ. The Gαs subunit produces the activation of the second messenger adenylate cyclase (AC) which catalyzes the conversion of ATP into cyclic adenosine monophosphate cytoplasmic (cAMP) by increasing the level of cAMP and the activation of protein kinase A (PKA), while the Gαi subunit stimulates the release of [3 H] arachidonic acid (AA) and induces epidermal growth factor receptor (EGFR) transactivation and subsequent DNA synthesis and inhibition of AC. Other effectors of the Gαq/_11_ pathway are phospholipase C-β (PLCβ), which stimulates the hydrolysis of phosphatidylinositol (PI) the formation of IP3 (IP3), and diacylglycerol (DAG) eliciting the release of Ca^2+^. Also, DAG diffuses along the plasma membrane where it may activate a ser/thr kinase called protein kinase C (PKC) which is also activated by an increase in the intracellular level of Ca^2+^. Moreover, Gαo is also crucial for the activation of the Wnt-β-catenin signaling pathway; Gα_12/13_ regulates changes in cytoskeletal pathway Rho [Encyclopedia of Signaling Molecules Second Edition]. On the other hand, the Gβγ subunit activates effectors such as PLCβ, AC, PI3K, K+ ion channels, and Src [21]. The NK-1R activates members of the mitogen-activated protein kinase (MAPK) cascade, including extracellular signal-regulated kinases 1 and 2 (ERK1/2), also called p42/44 MAPK, and several signaling pathways. One of these pathways depends on β-arrestin-mediated receptor endocytosis, while another pathway depends on tyrosine kinases or c-Src and transactivation of receptor tyrosine kinases. This suggests that this complex (β-arrestin and c-Src) might serve to link the receptor to the ras-dependent pathway of ERK1/2 activation mediated through the Gβγ subunit which is independent of protein kinase C that utilizes the Gαq/_11_ subunit [22] (Figure 1).

The NK-1R has two isoforms: a full-length 407 amino acid isoform NK-1R-Fl and a truncated isoform NK-1R-Tr (311 amino acids; because 96 residues are lost at the C-terminus) [23]. Due to the different structures of the isoforms, they have a different functional significance, differing in cell signaling capability [24]. The NK-1R-Fl exhibits a 10-fold higher binding affinity for SP than the NK-1R-Tr isoform [25]. It has been reported that all glioma cell lines, LN71, LN229, LN319, LN405, expressed NK-1R, however, the two NK-1R isoforms NK-1R-Fl and NK-1R-Tr are expressed in different concentrations. Only the LN319 glioma cell line presents the highest concentration of the NK1R-Fl isoform. This could have therapeutic implications [26]. In contrast, it has been reported that β-arrestin deficiency increases the sensitivity of glioma cells to treatment with NK-1R antagonists and is essential for NK1-mediated glioblastoma cell proliferation [27]. Moreover, it has been shown that the U-373 MG, U-87 MG, and GAMG glioma cell lines express both the NK-1R-Fl and NK-1R-Tr isoforms [10,28] and that, regardless of the amount of expression of the two isoforms, NK-1R antagonists, for instance L-733,060, L-732,138, and aprepitant, in a concentration-dependent manner, have the same therapeutic effect on GAMG, U-373 MG, and U-87 MG glioma cells, inhibit mitogenesis (as much as 100% inhibition at certain concentrations), and induce apoptosis of glioma cells [29,30,31,32]. Therefore, from a therapeutic point of view, the different expressions of the NK-1R-Fl and NK-1R-Tr isoforms in gliomas have no therapeutic implications. Importantly, NK-1R is essential for glioma viability of GAMG glioma cells [10]. Furthermore, in glioma samples, also called glioblastoma (grade 4 astrocytoma), all samples studied (10 of 10 [5] and 17 of 17 [6]) express NK-1Rs and, in the case of astrocytoma samples, 9 of 12; also, NK-1R density was more marked and more extensive in gliomas than in astrocytomas [5,7]. Glioma cell lines express NK-1Rs (e.g., LN71, LN229, LN319, LN405, U-373 MG, U-251 MG, DBTRG-05 MG, UC-11 MG, SNB-19, GAMG, U-87 MG) [10,16,26,28,30,33,34,35,36,37]. Moreover, it has been reported that glioma cells express 40,000–60,000 NK-1Rs per cell in UC-11 MG glioma [37]. It is known that the number of NK-1Rs is lower in cultures of glioma cell lines than in glioma tumors [5] and that poor prognosis and advanced tumor stages are related to the most malignant phenotypes overexpressing NK-1Rs [38]. It has been suggested that the degree of malignancy of a tumor correlates with the number of NK-1Rs: the greater the number, the higher is malignancy [18]. In glioma samples, it has also been reported that the number of these receptors was higher in tumors than in normal surrounding brain and that NK-1Rs were also found in blood vessels located within the tumor [5,7]. Thus, the overexpression of NK-1Rs in glioma could possibly be very useful for therapeutic intervention using NK-1R antagonists. 

## 3. SP Is Involved in Many Types of Signaling in Glioma

It is known that SP is located in the cytoplasm and the nucleus of glioma cells [10], indicating potential autocrine, paracrine, intracrine, and/or endocrine actions [39]. Autocrine function: cancer cells synthesize and release SP into the extracellular space by binding to the tumor cell’s own NK-1Rs, producing both glioma mitogenesis and antiapoptotic effects. Paracrine function: glioma cells synthesize and release SP into the extracellular space by binding to NK-1Rs from other glioma cells, endothelial cells, and inflammatory cells. As a result, it produces certain types of actions such as increased mitogenesis of glioma cells and endothelial cells and an antiapoptotic effect on glioma cells and stimulation of secretion of cytokines such as IL-1β, IL-6, IL-8, TNF-α, granulocyte macrophage colony-stimulating factor (GM-CSF), and leukemia inhibitory factor (LIF) in inflammatory and gliomas cells [33,40,41]. Intracrine function: glioma cells synthesize SP in the cytoplasm, which in turn can migrate to the nucleus and play a nuclear regulatory role in glioma cells. TATA box binding protein (TBP), a transcription factor in the minor groove of DNA, is known to lead to DNA deformation and DNA unfolding is facilitated by the overhang of two phenylalanine residues of TBP triggering mitogenesis in glioma cells [39,42]. Similarly, SP contains two phenylalanine residues at its C-terminus [14], being able to produce large-scale deformations in DNA and initiating transcription similar to that produced by TBP, producing mitogenesis in glioma cells, so essentially SP can be considered a transcription factor. Endocrine function: glioma tumor is composed of glioma cells which synthesize SP in the cytoplasm and release SP that can be released into the bloodstream and bind to NK-1Rs that are expressed in different cells throughout the body. Moreover, SP plasma concentrations were higher in cancer patients than in healthy subjects [19] (Table 1).

## 4. Role of SP and NK-1R in Mitogenesis and Antiapoptotic Effect in Glioma Cells

One of the hallmarks of cancer is the uncontrolled increase in mitogenesis in tumor cells. Furthermore, it is well known that SP is a universal mitogen in tumor cells [9,11,43,44]. There are numerous studies showing that SP after binding to NK-1R in glioma cells induces mitogenesis of glioma cells [29,45,46,47,48]. Moreover, it has been reported that SP binds to NK-1R of U-373 MG glioma cells, producing: a rapid increase in the synthesis of IP3, an increase in the concentration of cytosolic Ca^2+^ and the induction of immediate early transcription of the expression of the c-Fos and c-Jun (activator protein 1 (AP-1), a transcription factor) genes, and an increase in DNA synthesis [45]. SP activates NK-1R and induces the incorporation of [^3^H] thymidine into DNA and SP potently induces c-Myc mRNA and c-Myc protein expression. These genes are required for cell progression from the G1 phase to the S phase of the cell cycle in U-373 MG glioma cells. SP also stimulates NK-1R and activates tyrosine phosphorylation and enzymatic activity of MAPKs ERK1/2 [46]. SP activates phosphorylation of PHAS-I protein (also known as 4E-BP1) and p70S6 kinase (p70(S6K)), which is inhibited by rapamycin. In contrast, rapamycin does not suppress SP-induced activation of MAPK, c-Fos, and DNA synthesis in U-373 MG astrocytoma cells [49]. p70S6K is member of the serine/threonine protein kinase family, and it is one of the downstream effectors of the PI3K/Akt/mTOR signal transduction pathway. It phosphorylates S6 protein of the 40S ribosomal subunit and thus functions in protein synthesis and cell growth. It has been suggested that inhibition of the PI3K/Akt/mTOR pathway is an effective therapeutic strategy in tumors overexpressing phospho-p70S6K [50]. Similarly, it has also been suggested the PI3K/Akt/mTOR pathway is an effective therapeutic strategy in glioma [51]. Moreover, in U-373 MG glioma cells SP induced tyrosine phosphorylation of several proteins including EGFR and activated the EGFR complex containing the adapter proteins Sos, Shc, and Grb2, but not c-Src. SP activates the MAPK pathway, increasing ERK2 kinase activity and DNA synthesis in glioma cells, and EGFR is an essential regulator in SP/NK-1R-induced activation of the MAPK pathway and cell proliferation in U-373 MG glioma cells, and these events are mediated by subunit Gα_i_ protein [47] (Figure 1). However, it was later reported that blockade of the EGFR, which is transactivated by NK-1R, has minimal effect on NK-1R-mediated ERK1/2 phosphorylation. NK-1R-mediated ERK1/2 phosphorylation is significantly reduced by the c-Src kinase inhibitor PP2. Interestingly, ERK1/2 in U-373 MG glioma cells is also activated by several other mitogenic GPCRs, including alpha(1B)-adrenergic, M(3)-muscarinic, and H(1)-histaminergic in a c-Src-dependent manner. Thus, c-Src is a mediator of SP-stimulated ERK1/2 phosphorylation in human U-373 MG glioma cells [48]. In addition, EGFR transactivation by the NK-1R is only marginally involved in SP-dependent ERK1/2 phosphorylation because Src is a major component of NK-1R signaling, and the NK-1R also stimulates PKC δ phosphorylation [52]. Furthermore, the β-arrestin-mediated signaling pathway is essential for NK-1R-mediated glioma cell proliferation. β-arrestin knockdown inhibits NK-1R-mediated glioma cell proliferation and induces G2/M phase cell cycle arrest. β-arrestin knockdown cells showed remarkable downregulation of CDC25C/CDK1/cyclin B1 activity. β-arrestin-mediated ERK1/2 and Akt phosphorylation regulates the transcriptional activity of both NF-κB and AP-1, which were involved in cyclin B1 expression [27]. CDC25C/CDK1/cyclin B1 is a cyclin-dependent protein kinase that controls the cell cycle progression from G2 to M phase. It has been suggested that inhibition of the G1 and G2 phases of the cell cycle progression is an effective therapeutic strategy in tumors overexpressing CDC25/CDK1 induced by NK-1R activation [50]. Akt, or protein kinase B (PKB), is a serine–threonine protein kinase which becomes activated via phosphatidyl-3-kinase (PI3K) enzyme and suppresses apoptosis. In addition, SP binds to NK-1R-mediated Akt phosphorylation and the activation occurs through Src and PI3K but only partially through the EGFR kinase. NK-1R activates Akt through both EGFR-dependent and EGFR-independent pathways. This action of SP is counteracted by NK-1R antagonists L-733,060 and L-732,138 [30]. Another effect of Akt is the inactivation of the tumor suppressor gene p53, preventing its activity as a transcription factor of proapoptotic genes, decreasing its levels [53]. Once activated, Akt translocates to the various subcellular compartments where it phosphorylates several targets, including GSK-3β, and the phosphorylation leads to the inactivation of GSK-3β, and it creates a complex intracellular network [54] (Figure 1). In addition, it has been reported that the majority of DIPG tumors were positive for EGFR and p53, suggesting that deregulation of EGFR and p53 may play an important role in the development of DIPG and therefore therapies targeting these proteins might be beneficial in the treatment of DIPG [55].

## 5. Warburg Effect

Tumor cells are characterized by the production of hyperglycolysis followed by lactic fermentation, a phenomenon known as the Warburg effect [56]. Therefore, cancer cells need to obtain a large amount of glucose for hyperglycolysis. It is known that SP promotes the degradation of glycogen into glucose and increases the intracellular concentration of Ca^2+^ in glioma cells overexpressing NK-1R [57]. In contrast, these effects are completely blocked by the NK-1R antagonist CP-96,345 in a concentration-dependent manner [57]. Therefore, without glucose there can be no hyperglycolysis, leading to starvation of glioma cells. Moreover, it has been suggested that such effects are mediated by the NK-1R and that the glycolytic function is directly related to the number of NK-1Rs expressed in each cell, with tumor cells exhibiting higher expression levels and increased glycolytic rates [44]. Moreover, SP induces phosphorylation of GSK-3α/β (renders it inactive) in glioma cells [30]. In contrast, both NK-1R antagonists L-733,060 and L-732,138 counteract phosphorylation of GSK-3α/β in glioma cells [30]. Indeed, GSK-3 inhibitors reduce glucose production and increase glycogen synthesis from L-glucose. Therefore, GSK-3 inhibitors may reduce glucose levels [58]. The anti-Warburg effect of NK-1R antagonists is particularly important, as currently only this type of drug specifically counteracts the Warburg effect by preventing glucose formation in cancer therapy [44]. GSK-3 is activated by the canonical Wnt/β-catenin pathway and inhibits autophagy and the production of the autophagic adaptor p62. However, nutrient deprivation (such as glucose starvation) promotes the destruction of β-catenin through the inactivation of the Wnt pathway and Dvl is destroyed [51]. In summary, SP binds to the NK-1R of glioma cells and induces GSK-3α/β phosphorylation, glycogen degradation, and glucose formation. In contrast, NK-1R antagonists decrease both GSK-3α/β phosphorylation and glucose formation, counteracting the Warburg effect and inducing autophagy in glioma cells.

## 6. SP/HK-1 and NK-1R in Glioma Angiogenesis

Angiogenesis is a sequential process, in which there is early proliferation of endothelial cells and subsequent formation of new blood vessels, resulting in increased blood flow, accompanied by maturation of endogenous neurovascular regulatory systems, which occurs late in this process in inflamed tissues [11]. Moreover, angiogenesis is a hallmark of tumor development, associated with both increased tissue innervation and NK-1R expression [44]. Furthermore, NK-1Rs are found in intra- and peritumoral blood vessels. SP, through the NK-1R, which is found in high density in vessels, can strongly influence vascular structure and function in and around tumors by increasing tumor blood flow, which promotes, for example, stromal development and facilitates metastatic spread [5]. SP is expressed in brain capillary endothelial cells and is secreted by these cells in response to treatment with high doses of cytokines such as IL-1β and TNF-α [43,59]. In addition, NK-1R agonists, such as SP, can directly stimulate the process of neovascularization through induction of endothelial cell proliferation [60], and SP-enhanced angiogenesis results from a direct action on microvascular NK-1Rs. Subsequently it has been shown that endothelial cells express NK-1R and NK-2R [61]. In addition, vascular endothelial growth factor (VEGF) is a growth factor that promotes angiogenesis, has a mitogenic and antiapoptotic effect on endothelial cells, increases vascular permeability, and promotes cell migration [62]. Overexpression of VEGF in tumors is associated with increased angiogenesis, proliferation, and metastasis [63]. Furthermore, HK-1 has been reported to be a key endogenous regulator of angiogenesis. HK-1, after binding to NK-1R of human umbilical vein endothelial cells (HUVECs), concentration-dependently stimulated proliferation, migration, adhesion, and tube formation and also had angiogenic effects in vivo. In contrast, the angiogenic effects of HK-1 were inhibited by the selective NK-1R antagonist. In addition, HKs activated ERK1/2 phosphorylation, stimulated nitric oxide production, and increased the expression of endothelial nitric oxide synthase (eNOS) and VEGF in HUVECs [64]. VEGF overexpression plays an essential role in glioma progression and indicates poor glioma outcome, so VEGF may be an important prognostic factor in the overall survival of glioma patients [65]. In summary, glioma cells express SP and HK-1 [10,16] and, upon release, SP and HK-1 bind to the NK-1R of glioma vessel endothelial cells, inducing endothelial cell proliferation, migration, adhesion, tube formation, and activation of ERK1/2 phosphorylation leads to stimulation of nitric oxide production and increases in eNOS and VEGF expression which are involved in glioma angiogenesis.

## 7. SP and NK-1R in Inflammation and Microenvironment of Glioma

In 1863, Rudolf Virchow observed the presence of leukocytes in neoplastic tissues and established a connection between inflammation and cancer. Accordingly, it has been suggested that inflammatory cells and cytokines found and released in tumors contribute more to tumor growth, progression, and immunosuppression than to generating an effective antitumor response from the host. Therefore, the use of cytokine and chemokine blockers and non-steroidal anti-inflammatory drugs could be useful in the chemoprevention and treatment of malignant diseases [66]. SP is a main mediator of neuroimmunomodulatory activities and neurogenic inflammation within the central and peripheral nervous system. Neurogenic inflammation, which is characterized by vasodilation, increased permeability of postcapillary venules, plasma leakage, leukocyte infiltration, and adherence of neutrophils to blood vessels [41] and the release of neuropeptides including SP and calcitonin gene related peptide (CGRP). Inflammatory and glioma cells express and can release SP that regulates neurogenic inflammation mediated by the release of IL-1β, IL-6, IL-8, TNF-α, GM-CSF, and LIF [33,40,41]. It is known that human astrocytic and microglial cells constitutively overexpress the NK-1R-Fl isoform but not NK-1R-Tr. Furthermore, SP can enhance inflammatory and/or neurotoxic immune responses of glial cells to different inflammatory processes. So, SP, by binding to NK-1R of glial cells, elicits nuclear translocation of NF-κB [67]. Furthermore, microglia are known to secrete numerous inflammatory mediators, such as tumor necrosis factor α (TNF-α,) IL-1β, IL-6, and monocyte chemoattractant protein 1 (MCP-1) [68]. SP activates NF-κB, a transcription factor involved in the control of cytokine expression and apoptosis. SP increased the expression and secretion of interleukin-8 (IL-8), a target gene controlled by NF-κB, and this effect is specific, since an NK-1R antagonist completely prevented NF-κB activation in response to SP, but not to IL-1β, in U-373 MG glioma cells [69]. SP and histamine also induce interleukin-6 expression by a mechanism involving PKC and nuclear factor-IL-6 in U-373 MG glioma cells [70]. Furthermore, SP also induces phosphorylation of p38 MAPK which mediates SP-induced IL-6 expression in U-373 MG glioma cells by a mechanism independent of ERK1/2 (p42/44 MAPK), PKC, and NF-κB activation [71] (Figure 1). However, ketamine, a parenteral anesthetic drug, acts as a competitive antagonist of the excitatory neurotransmitter N-methyl-D-aspartate receptor and antagonizes SP functions by binding to the NK-1R of glioma cells by inhibiting the synthesis of IL-6 and IL-8 in U-373 MG glioma cells through the suppression of the phosphorylation of several signaling molecules such as NF-κB, p38MAPK, and p42/44 MAPK [72]. SP is involved in inflammatory conditions; in fact, the human plasma levels of SP are increased under certain pathologic conditions, including HIV infection [73] and cancer [19]. Furthermore, elevated circulating SP levels are associated with decreased numbers of natural killer (NK) cells and impaired function. SP inhibits the cytotoxic capacity and reduces NK degranulation. NK cells express NK-1R and this inhibitory effect of SP on cytotoxicity was partially prevented by the NK-1R antagonist CP-96,345. Thus, SP regulates NK cell functions and acts downstream of NK-1Rs to modulate NK cell activation signaling [74]. This mechanism may contribute to the impaired NK cell function in cancer patients in whom circulating SP is increased [19]. In contrast, using a selective NK-1R antagonist increases NK functions in cancer patients [74]. In addition, NK cells have been reported to lyse all DIPG cell cultures, suggesting that NK cells may represent a promising avenue for targeting DIPG [75]. Macrophages are components of the tumor microenvironment. Macrophages can differentiate along a spectrum of phenotypes, with M1 and M2 being the extremes. M1 macrophages are considered antitumoral and M2 macrophages are predominantly expressed within the tumor microenvironment and are considered protumoral. SP can induce M2 polarization of inflammatory macrophages. SP induces differentiation of GM-CSF-differentiated proinflammatory macrophages into M2-type alternatively activated phagocytic macrophages (M2SP) by direct activation of the PI3K/Akt/mTOR/S6kinase pathway and induction of arginase-1, CD163, and CD206, whereas treatment with the NK-1R antagonist RP67580 prevents all these processes [76]. However, it has been reported that DIPG tumors do not have a highly immunosuppressive or inflammatory microenvironment. Therefore, a potential immunotherapeutic should target the recruitment, activation, and retention of tumor-specific immune effector cells, such as NK cells [75]. Mast cells are another component of the tumor microenvironment. SP at nanomolar concentrations induces gene expression and secretion of VEGF in human mast cells and cultured mast cells derived from human umbilical cord blood (hCBMC). This effect is enhanced by co-administration of IL-33 in both cell types. In contrast, this effect of SP on VEGF release is inhibited by treatment with the NK-1R antagonist L-733,060. Although the PKC pathway is important for induction of VEGF by SP, this is not obligatory for its maximum induction. Also, SP stimulates phosphorylation of the MAPKs ERK and JNK, which can be activated by PKC-dependent and PKC-independent mechanisms. Activation of these MAPKs leads to activation of the transcription factor AP-1, a heterodimer of c-Fos and c-Jun. The VEGF promoter has several AP-1-binding sites that increase transcription, which may explain the increased abundance of VEGF mRNA in SP-stimulated cells. These results imply that functional interactions between SP, IL-33, and mast cells leading to VEGF release contribute to inflammatory conditions [77] (Figure 1). In addition, it has been reported that the inhibition of NK-1R expression can decrease the phosphorylation levels of MAPKs (ERK1/2, JNK, and p38 MAPK) in RBL-2H3 mast cells [78]. Since SP is a mediator of neurogenic inflammation, SP induces prostaglandin (PG) production in various cell types, and these eicosanoids are responsible for numerous inflammatory and vascular effects. Cyclooxygenase (COX) is needed to convert arachidonic acid to PGs. COX-2 protein expression is upregulated by SP and thus there exists a correlation between COX-2 expression and PGI (2) and PGE (2) release. Dexamethasone (DEX) inhibited SP-mediated COX-2 expression. MAPK p38 and ERKs (p42/44 MAPK) were activated by SP. NK-1R and NK-2R but not NK-3Rs are present on HUVECs. Selective NK-1R and NK-2R agonists upregulated COX-2 protein expression and PG production. The NK-1R antagonist L-703,606 and the NK-2R antagonist SR 48,968 competitively antagonized SP-induced effects [61]. SP can initiate angiogenesis during acute neurogenic inflammation by increasing the proliferation rate of endothelial cells. In contrast, administration of an NK-1R antagonist, SR140333 (nolpitantium), prevented this phenomenon [79]. In addition, the growth of new vessels from pre-existing vasculature is a common feature of chronic inflammation, and early angiogenesis is a key step in the transition from acute to persistent inflammation [44]. In summary, the tumor inflammatory microenvironment is composed of glioma cells, inflammatory cells, and endothelial cells, all express NK-1R and SP, so they all release SP that, when binding to the NK-1R of the cells of the tumor microenvironment, releases different inflammatory interleukins, producing two negative effects, stimulation of tumor growth and depressing the patient’s immunity. Therefore, the use of the NK-1R antagonist would counteract the pathological effects mediated by SP, decreasing inflammation and tumor growth and improving the patient’s immunity.

## 8. SP/HK-1 and NK-1R in Migration and Invasion of Glioma Cells

The migration of tumor cells is the previous step for the invasion and dissemination of primary solid tumors, causing infiltration and secondary metastases in distant organs. It is known that tumor cell migration is induced by classical neurotransmitters (dopamine, noradrenaline) and peptides such as SP [80].

### 8.1. Epithelial–Mesenchymal Transition (EMT)

EMT is a biological process that allows a polarized epithelial cell to undergo multiple biochemical changes that allow it to assume a mesenchymal cell phenotype, which includes greater migratory capacity, invasiveness, high resistance to apoptosis, and greatly increased production of ECM components [81]. In general, in tumors, EMT contributes to initiation migration, by loss of cell–cell adhesion, invasion by acquisition of migratory and metastasis to distant sites by loss of cell polarity, and resistance to chemotherapy and/or radiotherapy. In adults, intrinsic chemoresistance to glioma has been linked to EMT. Additionally, pediatric high-grade glioma (HGG) and DIPG are extremely chemoresistant, partially due to the blood–brain barrier (BBB) and due to intrinsic factors [82]. EMT has also been suggested to play an important, yet poorly understood, role in DIPG biology and therapy resistance [82]. EMT is induced by external factors, such as cytokines, growth factors, and hypoxia. The activation of EMT transcription factors is produced by different signaling pathways, most importantly SNAI1 (SNAIL), SNAI2 (SLUG), ZEB1, ZEB2, and TWIST. These transcription factors inhibit the expression of genes that promote cell–cell contact, most importantly E-cadherin, and induce genes that promote migration and stemness [82]. Cadherin is a transmembrane glycoprotein involved in intercellular adhesions and signaling, and the cadherin switch is the result of activation of the EMT-related signaling pathways. E-cadherin (CDH1) is expressed in epithelial tissues and N-cadherin (CDH2) is found in neural tissue, muscle, and fibroblasts. EMT transcription factors, after binding to cadherins, cause the loss of expression of E-cadherin and induction of expression of N-cadherin, triggering a series of events that allows cancer cells to become less dependent of their microenvironment, thereby promoting cell migration, invasion, and metastasis in epithelial tumor cells [83]. This possibility of a switch from E- to N-cadherin and its relationship to EMT are unsettled. There has been a report demonstrating double labeling for E- and N-cadherin in MPAP (invasive micropapillary carcinomas of breast cancer) tumors. This report revealed that, within individual tumors, N- and E-cadherin expression can be heterogeneous, suggesting the possibility that the tumor cells first downregulate E-cadherin in order to invade and then can re-express E-cadherin as the tumor develops [84]. Several studies have also shown that changes in N-cadherin levels occur in malignant gliomas [85]. Interestingly, it is known that SP induces EMT and matrix metalloproteases that promote tumor invasion in head and neck cancer (HNC) cells [86]. In contrast, NK-1R antagonist L-703,606 counteracts several of these upregulated EMT-associated genes, producing antimetastatic effects [86]. In addition, the NK-1R antagonist aprepitant has been reported to counteract the progression of EMT in a Balb/C mouse model of endometriosis [87]. Thus, the use of NK-1R antagonists such as the drug aprepitant may counteract the progression of EMT which may play an important, yet poorly understood, role in DIPG biology and therapy resistance.

Activation of EMT by the Wnt/β-catenin pathway allows cancer cells to survive individually, invade surrounding tissues, and metastasize [51]. It is known that self-renewal, proliferation, and differentiation of neural progenitor cells (NPCs) in the brain during different stages of the central nervous system (CNS) are regulated by Wnt signaling [51]. Wnt signaling can be subdivided into canonical and non-canonical groups (β-catenin independent). Glioma and other cancers have been linked to abnormal Wnt pathway activity; aberrant activation of Wnt/β-catenin signaling may play a crucial role in tumorigenesis. Importantly, active Wnt/β-catenin signaling has been associated with decreased glioma patient survival [88]. In the absence of Wnt, cytoplasmic β-catenin is constantly degraded by the Axin complex, composed of the Axin protein, the adenomatous polyposis coli (APC) tumor suppressor gene, casein kinase 1 (CK1), and GSK-3. Meanwhile, Wnt ligands bind to the transmembrane receptor frizzled (Frz) and the co-receptor lipoprotein-related proteins 5 and 6 (LRP-5/6) inactivate GSK-3β and prevent it from phosphorylating β-catenin, thus stabilizing β-catenin in the cytoplasm. As β-catenin accumulates, it translocates into the nucleus where it binds to TCF/LEF and increases their transcriptional activity. Genes upregulated by TCF/LEF include proto-oncogenes, such as c-Myc and cyclin-D1, and genes regulating cell invasion/migration, such as MMP-7 [89]. Wnt proteins are also able to regulate the β-catenin-independent pathways, so-called non-canonical Wnt pathways that activate RhoA-Rock, Rac-1-JNKs, calcium/calmodulin-dependent protein kinase II, PKC, and phospholipase C, for cell motility and polarity [90]. It has been reported that SP increases the expression of the genes and proteins associated with activation of the Wnt/β-catenin signaling pathway of bone marrow stromal stem cells (BMSCs), so SP increases the proliferation of stem cells [91]. Moreover, Wnt signaling is involved in SP’s inhibition of apoptosis. SP exerts a protective effect, reduces the apoptotic rate, reduces nuclear condensation, inhibits the activation of caspase-3 and caspase-9, and reduces the percentage of cleaved caspase-3-positive cells. SP promotes the mRNA and protein expression of Wnt signaling molecules such as β-catenin, GSK-3β, c-Myc, and cyclin D1 in bone marrow stromal stem cells [92]. Furthermore, the increased transcriptional activity of β-catenin results from the activation of EGFR through ERK that directly interacts with casein kinase 2 (CK2) without altering its stability and the level of phosphorylation by GSK-3β in human glioblastoma cells U-87E, U-373, and LN229 and correlates with levels of ERK1/2 activity and grades of glioma malignancy [93] (Figure 1).

### 8.2. Blebbing on Glioma Cell Membranes

Blebbing on tumor cell membranes is important in the spread and invasion of cancer cells [94]. SP, after binding to NK-1Rs, induces changes in cellular cancer shape, including blebbing, which is crucial in cell movement/spreading and in cancer cell infiltration [95]. Moreover, it has been reported that Rho-associated protein kinase (Rock) is also involved in these changes, and in glioma cells SP induced the phosphorylation of p21-activated kinase (PAK) and enhanced phosphorylation of myosin regulatory light chain kinase (MLCK), resulting in bleb formation; however, this phenomenon was not observed in normal human cells [96]. In non-canonical Wnt/PCP signaling, the interaction of Wnt/Fzd leads to the recruitment of Dvl, which utilizes its domains (PDZ and DIX) to produce a complex with DAAM and then stimulates RhoA which can activate Rock [51]. Therefore, SP produces blebbing via non-canonical Wnt/PCP signaling that is mediated by subunit Gα_0_ and via subunit Gα_12/13_ [21] (Figure 1).

### 8.3. The Extracellular Matrix (ECM)

ECM is the non-cellular component of tissues, which has been compared to the “glue” that holds cells together in connective tissues, where it is an important component of the tissue [97]. Matrix metalloproteinases (MMPs) are a group of active protein hydrolases found in the extracellular matrix. MMPs can degrade the ECM, leading to tumor invasion and metastasis. The overexpression and secretion of MMP-2 and MMP-9 is increased in several types of human cancers and is associated with poor prognosis. Increased MMP-2 and MMP-9 expression has also been reported to correlate with cancer invasion [98]. MMP-2 and MMP-9 have also been closely associated with glioma progression and malignancy [99]. In glioma, it is known that HK-1, through NK-1R, dose-dependently promotes the migration of U-251 and U-87 glioma cells. Furthermore, HK-1 increased MMP-2 activity, as well as the expression of MMP-2 and MT1-MMP (activator of MMP-2), which regulate cell migration. In U-25 glioma cells, HK-1 by NK-1R binding induces an increase in calcium release. However, inhibiting PLC reduces calcium release, MMP-2 and MT1-MMP activation, and HK-1-induced cell migration, suggesting that the NK-1R-mediated migration effect is by the Gαq-PLC pathway. HK-1 also increases phosphorylation of ERK, JNK, and Akt and inhibition of ERK and Akt decreases the induction of MMP-2 and MT1-MMP by HK-1. Furthermore, HK-1 phosphorylates p65 and c-Jun and increases both AP-1 and NF-kB activity in U-251 glioma cells [31] (Figure 1).

In summary, SP/HK-1 by NK-1R causes strong regulation and progression of EMT-associated genes in tumor cells. SP also induces blebbing of glioma cell membranes and increases the expression of MMP-2 and MT1-MMP in glioma cells, resulting in migration and invasion of human glioma cells.

## 9. NK-1R Antagonists

It is known that NK-1R antagonists, after binding to the NK-1R and in a concentration-dependent manner, counteract all SP-mediated pathophysiological actions. NK-1Rs constitute a heterogeneous group of compounds. There are two types of NK-1R antagonists: non-peptide antagonists and peptide antagonists.

Peptide NK-1R antagonists (in the SP molecule, L-amino acids are changed to D-amino acids) are also known as SP analogue antagonists, synthetic analogues of SP, or broad-spectrum peptide antagonists. Peptide antagonists of NK-1R show lower affinity for NK-1R than natural peptide agonists; they are metabolically unstable because they are hydrolyzed by peptidases; they are not lipid soluble so they do not penetrate the brain; and some of them have toxic effects; therefore, they cannot be used for the treatment of glioma [8,44].

Non-peptide NK-1R antagonists include compounds that exhibit similar stereochemical characteristics and different chemical compositions. The therapeutic activity is linked to both the affinity of these compounds for the NK-1R and the concentration of the compounds [8,44]. Non-peptide antagonists of the NK-1R are not degraded by peptidases and are lipid soluble/brain penetrants [8,44]. Currently, there are the following NK-1R antagonist drugs: aprepitant, fosaprepitant, netupitant, fosnetupitant, rolapitant, tradipitant [100]. Unfortunately, the NK-1R antagonist drugs netupitant, fosnetupitant, rolapitant, and tradipitant are currently only known for their antinausea and antivomiting activities and their antitumor activity in gliomas has not been studied to date.

## 10. NK-1R Antagonists in Glioma Therapy

### 10.1. Brain Penetrant

The BBB prevents the use of most antitumor drugs, as they cannot cross it. Therefore, it is essential that the drugs used can cross the BBB in the treatment of glioma. It is well known that the NK-1 receptor antagonist aprepitant is brain penetrant. In fact, aprepitant is used in the treatment of chemotherapy-induced nausea and vomiting (CINV).

### 10.2. Antitumor Action of NK-1R Antagonists in Glioma

NK-1R antagonists CP-96,345, L-733,060, L-732,138, MEN-11,467, MEN-11,149, and the drugs aprepitant and cyclosporine A (CsA) are known to have antitumor activity in glioma. All of them produce inhibition of proliferation and death by apoptosis in glioma cells. In addition, it is known that inducing apoptosis in tumor cells is a key anticancer treatment strategy [101].

### 10.3. Justification for Use of NK-1R Antagonists in Glioma

#### NK-1R Antagonists Inhibit Mitogenesis and Induce Apoptosis in Glioma Cells

SP, via NK-1Rs of U-373 MG glioma cells, induces mitogenesis and induces the synthesis of c-Myc mRNA and protein, and SP activates the MAPKs ERK1/2 and p38MAPK. In contrast, NK-1R antagonist CP-96,345, in a concentration-dependent manner, suppresses DNA synthesis and activation of the MAPK pathway [46]. Furthermore, in a human glioma xenograft model, NK-1R antagonists (MEN-11,467 and MEN-11,149) have been shown to decrease tumor growth and exhibit antitumor activity [102]. In addition, three different NK-1R antagonists (L-733,060, aprepitant, L-732,138), at micromolar concentrations, inhibit the proliferation of GAMG glioma cells [29,32]. The antiproliferative action of the three NK-1R antagonists mentioned above is related to the affinity for NK-1R: the most potent antagonist against GAMG glioma cells was L-733,060 followed by aprepitant and L-732,138 [29,32]. The immunosuppressive drug CsA has also been shown to exert an antitumor action against GAMG glioma cells [103]. However, in clinical practice, the use of CsA is not possible, since the required antitumor dose induces nephrotoxicity. On the other hand, β-arrestin deficiency increases the sensitivity of U-251 MG and U-87 MG glioma cells to treatment with NK-1R antagonists L-732,138 and [D-Arg^1^, DTrp^5,7,9,^ Leu^11^] SP increases cell apoptosis and G2/M phase arrest [27].

GSK-3, a serine/threonine kinase, is involved in diverse cellular processes from nutrient and energy homeostasis to glioma cell proliferation and apoptosis. Inhibition of GSK-3 activity induces phosphorylation of c-Myc site S62 and results in increased expression of apoptosis-related Bax, Bim, DR4/DR5, and TRAIL expression and subsequent cytotoxicity and apoptosis of glioma cells. Thus, GSK-3 may therefore be an important therapeutic target for glioma treatment [104]. In addition, NK-1R antagonists L-733,060 and L-732,138, in a concentration-dependent manner, counteract SP-induced GSK-3 phosphorylation [30]. Moreover, SP phosphorylates Akt and exerts an antiapoptotic effect in glioma cells [30]. In contrast, NK-1R antagonist aprepitant has been shown to induce apoptosis in GAMG glioma cells [32]. Furthermore, the NK-1R antagonist L-733,060 via NK-1R inhibits the basal kinase activity of Akt; it increases apoptosis and results in caspase-3 cleavage and poly (ADP-ribose) polymerase proteolysis in U-373 MG glioma cells [30] (Figure 2). Moreover, basal Akt activity has been linked to poor prognosis in glioma [30]. It should be noted that NK-1R is essential for glioma cell viability [10]. Furthermore, it is known that all glioma cells overexpress NK-1R. That means that H3K27M-mutated diffuse midline glioma necessarily overexpresses NK-1Rs. Thus, the use of NK-1R antagonists such as aprepitant may be an effective therapy in the treatment of this type of glioma, producing inhibition of mitogenesis and death by apoptosis [30,32].

### 10.4. NK-1R Antagonists Mediate Antiangiogenic Effects

SP is a main mediator of neurogenic inflammation. SP and a selective peptide NK-IR agonist, [β-Ala^4^, Sar^9^, Met(O_2_)^11^]-SP (4–11), induce marked neovascularization. Moreover, SP through NK-1R of endothelial cells increases their proliferation in a concentration-dependent manner. Endothelial cell proliferation was followed by blood vessel formation and increased blood flow. Thus, SP can directedly stimulate the process of angiogenesis and neovascularization through induction of endothelial cells. In contrast, two NK-1R antagonists, [D-Pro^2^, D-Trp^7,9^]-SP and [D-Pro^4^, D-Trp^7,9,^ Ph^11^]-SP (4–11), counteract SP-mediated angiogenesis and proliferation of endothelial cells in a concentration-dependent manner [60]. NK-1R antagonists also inhibit the angiogenic effects of HK-1, such as nitric oxide production, eNOS, and VEGF expression in endothelial cells [64]. Furthermore, non-peptide NK-1 receptor antagonists, including the drug aprepitant, have been shown in xenograft models to block both angiogenesis and tumor growth [105,106]. NK-1R antagonist aprepitant also reduces the expression of VEGF and VEGFR in tumor cells [107,108,109] (Figure 2).

### 10.5. NK-1R Antagonists Mediate Antimetastatic Effects in Glioma

Metastasis causes 90% of cancer deaths [110]. Thus, one of the main goals of cancer treatment is to prevent this process. In this sense, it has been reported that NK-1R antagonist L-733,060 inhibited SP-mediated tumor cell migration in MDA-MB-468 breast and PC-3 prostate carcinoma cells [80]. Glioma cells can invade and spread to the brain parenchyma over long distances. This is not a metastatic process, but rather an active dissemination, either along myelinated axons or perivascular spaces, or through the cerebrospinal fluid. Therefore, the therapeutic goal of glioma invasion is to prevent the detachment of cancer cells from the initial tumor mass to limit its dissemination [85]. Furthermore, it has been reported that downregulation of N-cadherin in glioma cells resulted in cell polarization defects leading to abnormal motility behavior with increased cell velocity and decreased persistence of directionality. In contrast, re-expression of N-cadherin in glioma cells restored cell polarity and limited glioma cell migration, providing a potential therapeutic tool for diffuse glioma [85]. In addition, it has also been reported that targeted inhibition of ZEB2 induced upregulation of E-cadherin expression and downregulation of vimentin expression leading to reduced invasion and migration of pediatric glioma and adult glioma cells [111]. The cell adhesion molecule E-cadherin acts as an indispensable suppressor of cancer metastasis, and furthermore, increased E-cadherin can decrease β-catenin transcription, thereby decreasing Wnt/β-catenin signaling activity [107]. In colon cancer HCT116 cells, NK-1R antagonist NKP-608 reduced expressions of Wnt-3a, β-catenin, cyclin D1, and VEGF while inducing expression of E-cadherin. NK-1R antagonist NKP-608 also triggered apoptosis and induced apoptosis-related proteins reduced Bcl-2 and increased Bax and active caspase-3 [107]. Therefore, treatment with the NK-1 receptor antagonist aprepitant could increase N-cadherin in glioma cells, restore cell polarity, and limit glioma cell migration, similar to E-cadherin in colon cancer cells. Moreover, NK-1R antagonist aprepitant inhibited the expression of MMP-2, MMP-9, VEGF-A, and VEGF receptor1 (VEGFR1) in ovarian carcinoma cells and in osteosarcoma cells [108,109] (Figure 2). NK-1R antagonist aprepitant also inhibited SP-induced proliferation, cell migration, and invasion and induced apoptosis of human squamous cell carcinoma (ESCC) cells by downregulating the PI3K/AKT/mTOR signaling pathways [112].

In summary, therapeutic use of NK-1R antagonists such as aprepitant or similar drugs in DIPG patients can be useful because they are brain penetrant and reach therapeutic concentration in the central nervous system (CNS). The antitumor action of NK-1R antagonist drugs consists of inhibiting mitogenesis and promoting apoptosis of glioma cells, inhibiting angiogenesis by inhibiting both endothelial cell proliferation and VEGF, decreasing migration of tumor cells, counteracting blebbing formation, decreasing the expression of MMP-2 and MT1-MMP, and increasing E-cadherin in glioma cells, resulting in decreased migration and invasion (metastasis).

### 10.6. Anti-Inflammatory NK-1R Antagonists 

It has been reported that, in a model of inflammatory pain, NK-1R antagonist aprepitant suppresses microglial activation, JNK and p38 MAPK phosphorylation, and mRNA and protein expressions of TNF-α, IL-6, IL-1β, and MCP-1, in vitro and in vivo [68] (Figure 2).

Moreover, in an experimental brain tumor model, it has been reported that untreated animals showed increased albumin, SP, and NK-1Rs in the peritumoral area, as well as increased perivascular staining in the surrounding brain tissue. Brain water content and blood–brain barrier permeability were significantly increased in tumor-inoculated animals compared with controls. However, treatment with the NK-1R antagonist aprepitant reduced BBB dysfunction and edema formation. This suggests that SP plays a role in the genesis of peritumoral edema and, therefore, could be a possible antiedema treatment in brain tumors [113].

### 10.7. Combination of RT Plus NK-1R Antagonist

The current standard of treatment for DIPG consists of external beam RT at a dose of 54–60 Gy with conventional fractionation [114]. Radiotherapy produces a temporary improvement in neurological function, although the overall prognosis remains poor. Median overall survival is approximately 12 months after conventionally fractionated radiotherapy [115]. Unfortunately, it is well known that radiotherapy induces adverse effects on the surrounding normal brain, mainly producing inflammation and radionecrosis in the CNS [116]. The development of effective methods to enhance tumor radiosensitivity is crucial to improve the therapeutic efficacy of RT. Thus, there is a pressing need to find drugs that increase the antitumor activity of RT (radiosensitization) and reduce serious side effects such as inflammation and radionecrosis in DIPG. It has been reported that the combination of radiotherapy and NK-1R antagonist aprepitant exerted a synergistic antitumor activity and decreased the side effects promoted by radiotherapy [117]. In a lung cancer patient that was treated with palliative radiotherapy plus compassionate use of NK-1R antagonist aprepitant, after 45 days of treatment the lung tumor mass disappeared and the side effects of radiotherapy were not observed [117]. Furthermore, it is known that radiotherapy causes neurogenic inflammation through processes involving NK-1R in rats [118]. In contrast, pretreatment with NK-1R antagonists GR203040 or GR205171 (at a low dose of 0.3/mg/kg) or dexamethasone inhibited radiotherapy-induced plasma protein extravasation in the duodenum and jejunum; this inhibition was increased when the combination of an NK-1R antagonist plus dexamethasone was used [118]. Interestingly, NK-1R antagonists GR203040 and GR205171 penetrate the brain [119]. Therefore, NK-1R antagonists decrease neurogenic inflammation induced by radiotherapy [118]. In addition, the radioprotective potential of N-acetyl-L-tryptophan (L-NAT), an NK-1R antagonist, in neuronal cells has been shown [120]. Pretreatment with the NK-1R antagonist L-NAT resulted in DNA protection. NK-1R antagonist L-NAT pretreatment of irradiated Neuro2a cells neutralized oxidative stress and reduced mitochondrial dysfunction. The expression of caspase-3 and γ-H2aX proteins was decreased, while the expression of p-ERK1/2 and p53 was increased in irradiated cells pretreated with L-NAT compared with irradiated cells. Therefore, NK-1R antagonist L-NAT could be a potential radioprotective that can inhibit oxidative stress and DNA damage and maintain mitochondrial health and Ca2+ levels by activating the expression of p-ERK1/2 and p53 in neuronal cells [120]. In addition, combination therapy of chemotherapy or radiotherapy and an NK-1R antagonist as a new strategy in cancer therapy has been suggested in a recent report [121]. In summary, the combination of radiotherapy plus NK-1R antagonist aprepitant or similar drugs produces both radiosensitization of the tumor and can decrease the severe side effects of radiotherapy. Thus, this combination therapy could be useful in DIPG treatment.

### 10.8. SP/NK-1R System in Targeted Tumor Therapy with Radionuclides in DIPG

Overexpression of the NK-1R and the SP/NK-1R system has been linked to cancer cell progression and poor overall prognosis, including in DIPG. Based on the high density of transmembrane NK-1Rs that tumor cells expressed, several attempts have attempted to utilize peptide vectors based on endogenous NK-1R ligand, SP or its close analogues, and small molecular antagonists, NK-1R vectors, as radiopharmaceuticals in tumor therapy with radionuclides [6,12]. Recently, some novel structural modifications of L-732,138 and aprepitant have been developed to create novel radioconjugates with different radionuclide chelators. All of them were radiolabeled with ^68^ Ga and ^177^ Lu radionuclides and evaluated in terms of their lipophilicity and stability in human serum with the aim of providing potential theranostic-type radiopharmaceutical pairs for imaging and therapy of cancers with NK-1R overexpression [122,123]. However, L-732,138 containing DOTA chelator coupled to L-732,138 and its analogues labeled with ^68^ Ga and ^177^ Lu showed rather limited stability in human plasma, therefore application must be in locoregional administration [123]. More recently, new radioconjugates based on new homologues of aprepitant functionalized with the chelator DOTA and labeled with ^68^ Ga and ^177^ Lu have been synthesized and developed with a view to future oncological applications. These new radioconjugates showed a significant increase in lipophilicity compared to the previous ones and maintained stability in human serum. Regarding the binding to human NK-1R, the new radioconjugates showed a high affinity and greater binding capacity [124]. In summary, these articles suggest the possibility of a new treatment of the SP/NK1R system in targeted tumor therapy with radionuclides in DIPG.

### 10.9. Clinical Experience with the Drug Aprepitant, an NK-1R Antagonist

The clinical use of aprepitant for chemotherapy-induced nausea and vomiting (CINV) was approved twenty years ago. Therefore, it is a well-known drug in oncology and has widespread clinical use. The side effects of aprepitant and fosaprepitant (a prodrug of aprepitant for intravenous use) are minimal, the most common (incidence higher than 10%) of which are constipation, fatigue, headache, anorexia, hiccups, and diarrhea [125]. Chemotherapy has an extremely low therapeutic index (at therapeutic doses, chemotherapeutics are also cytotoxic to normal tissues). In contrast, the therapeutic index of aprepitant is high. Adverse events associated with aprepitant in pediatric bone cancer patients include febrile neutropenia in frequencies of over 43.8 per 100 patients in one study [126]. In contrast, in a meta-analysis of 23 randomized controlled trials with 7956 patients, aprepitant was shown to be safe with no statistically significant differences in the incidence of febrile neutropenia [127]. Moreover, the cytotoxicity effect of aprepitant has been shown to be eight-fold higher in cancer cells than in normal cells (lymphocytes) [128]. As bone marrow toxicity is a requirement for neutropenia and aprepitant does not produce bone marrow toxicity, there should be no reason why aprepitant would cause febrile neutropenia in the setting of cancer patients. Furthermore, it is well known that aprepitant at 300 mg/day was safe and well-tolerated and showed side effects similar to placebo [129]. Moreover, the use of a combination of palliative radiotherapy plus compassionate use of aprepitant for 45 days (1140 mg/day for 45 days) in a lung cancer patient was reported to generate very good clinical outcomes. Radiotherapy was administered to the right lung and mediastinum, reaching 50.4 Gy, and then overprinted to 65 Gy. In each successive control, the tumor volume decreased and, after 6 months of treatment, the chest computed tomography scan showed that the tumor mass had disappeared. No side effects were observed during the combination therapy; rather, the patient was in very good general health with weight gain and showed no biochemical analytical alteration [117]. Moreover, aprepitant may be useful in metastasis. It has been reported that a patient with brain metastasis from breast cancer refractory to standard antiemetic therapies, when treated with aprepitant 80 mg/day and subsequently 125 mg/3 days for 6 months, improved clinically and the tumor marker CA153 decreased from 187 to 122, and the patient also achieved control of nausea and vomiting without any side effects [130]. Administration of aprepitant (375 mg/day/2 weeks) to HIV patients was shown to be safe and well tolerated and resulted in a decrease in the number of CD4+ PD-1-positive cells, with a reduction in both soluble CD163 and plasma SP levels [131]. Furthermore, overexpression of CD163 in glioma samples correlated with poor patient prognosis. CD163 expression is increased in glioma cells, especially in primary glioma cells. Loss of CD163 expression inhibited both cell cycle progression and proliferation of glioma cell lines and primary glioma cells. CD163 directly interacted with casein kinase 2 (CK2), and silencing of CD163 reduced the activity of the AKT/GSK-3β/β-catenin/cyclin D1 pathway through CK2. Thus, CD163 contributes to gliomagenesis through CK2 and suggests that the CD163 pathway could serve as a therapeutic target for glioma [132].

The different doses of aprepitant have different therapeutic effects: aprepitant doses (first day 125 mg; second day 80 mg; third day 80 mg) for CINV. Aprepitant 300 mgday is used as an antidepressant [129] and aprepitant 375 mg/day is used as an anti-inflammatory in HIV patients [131], both are moderate doses. The suggested dose for the treatment of glioma is 20–40 mg/kg/day (based on extrapolation from the concentration used in preclinical studies) [9,32]. In 1907 (more than 100 years ago), German Nobel Prize winner Paul Ehrlich developed a revolutionary scientific concept in pharmacology, the magic bullet, which would be specific against tumor cells but not against the patient’s normal cells. NK-1R antagonist drugs are new and promising anticancer drugs and could be considered as a new magic bullet called an “intelligent bullet” [32]. The concept goes further because, in addition to their multiple antitumor actions, they have other therapeutic effects on patients such as control of CINV, antidepressant, anti-inflammatory, and immunoprotective, and they are also safe.

## 11. Conclusions

Current and generally non-specific antitumor treatments such as radiotherapy present two major problems, the resistance of glioma cells and serious side effects. For this reason, new antitumor procedures targeting glioma cells must be urgently developed; they must be effective drugs without serious side effects. Many preclinical studies have shown that the SP/NK-1R system is a good antitumor target because it is overexpressed in glioma cells and NK-1Rs are essential for the viability of glioma cells and not for normal cells. The SP/NK-1R system is involved in the proliferation of glioma cells, has antiapoptotic effects on glioma cells, induces the Warburg effect, triggers angiogenesis, and produces inflammation as well as the progression of EMT, thereby increasing the migration and invasion of glioma cells (infiltration and metastasis) and development of resistance to chemotherapy and/or radiotherapy. In contrast, NK-1R antagonists such as aprepitant or similar drugs penetrate the brain and, in a concentration-dependent manner (high doses), have many therapeutic effects on glioma, such as: inhibiting mitogenesis and inducing apoptosis in glioma cells, counteracting the Warburg effect in glioma cells, inhibiting angiogenesis, having anti-inflammatory effects, counteracting the progression of EMT by decreasing glioma cell migration and invasion (invasion and metastasis), as well as preventing induction of resistance to chemotherapy and/or radiotherapy. Furthermore, the combination of radiotherapy plus aprepitant may increase the antitumor activity of radiotherapy (radiosensitization of glioma). In addition, it has an anti-inflammatory effect both on peritumoral inflammation of the glioma tumor and on inflammation caused by radiotherapy and reduces the serious side effects of radiotherapy; it is neuroradioprotective by preventing radiotoxicity and radionecrosis. Moreover, aprepitant or similar drugs present other therapeutic effects in glioma patients such as antinausea and vomiting. Therefore, NK-1R antagonist drugs can be considered as a new magic bullet in the treatment of glioma, perhaps more accurately called a “intelligent bullet”, and we suggest the use of aprepitant to treat DIPG in combination with radiotherapy. In summary, the SP/NK-1R system provides a promising therapeutic target (NK-1R) in glioma and novel antitumor drugs (NK-1R antagonists) in glioma therapy. Therefore, aprepitant or similar drugs should be repurposed using NK-1R antagonist drugs in combination with radiotherapy in patients with DIPG. Furthermore, we suggest that phase I clinical trials are needed to evaluate the safety of aprepitant and phase II trials to evaluate its efficacy in combination with radiotherapy in patients with DIPG.

## Figures and Tables

**Figure 1 cancers-17-00520-f001:**
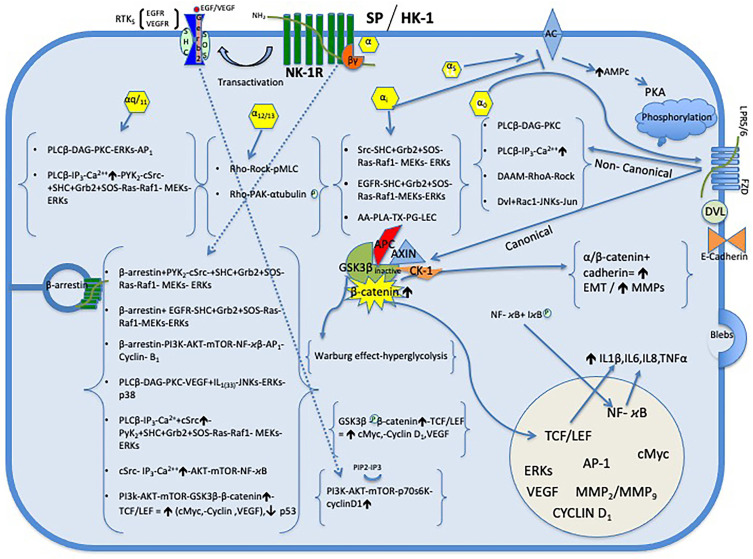
The substance P/neurokinin-1 receptor (SP/NK-1R) system regulates multiple cell signaling pathways involved in cancer progression: cell proliferation and migration, angiogenesis, inflammation, antiapoptotic mechanisms, Warburg effect, and in the suppression of the immune system. Abbreviations: AA, arachidonic acid; AC, adenylate cyclase; AKT or PKB, Protein kinase B; APC, adenomatous polyposis coli; CK-1, casein kinase 1; DAAM, disheveled-associated activator of morphogenesis; DAG, diacylglycerol; Dvl; disheveled; EGF, epidermal growth factor; EGFR, epidermal growth factor receptor; EMT, epithelial–mesenchymal transition; ERK1/2, extracellular signal-regulated protein kinase; Fzd, frizzled; GSK-3β, glycogen synthase kinase 3β; HK-1, hemokinin-1; IP3, inositol trisphosphate; LEC, leukotriene; LRP5/6, low-density lipoprotein receptor-related protein 5/6; MAPKs, mitogen-activated protein kinases; MEKs, mitogen-activated protein kinase kinases; MMPs, matrix metalloproteinases; NF-κB, nuclear factor kappa B; PKC, protein kinase C; PG, prostaglandin; pMLC, myosin light-chain kinase; PI3K, phosphoinositide 3-kinase; PLA, phospholipase A; PLC β, 1-phosphatidylinositol-4,5-bisphosphate phospholipase beta; ROCK, Rho-associated kinase; RTK, receptor tyrosine kinase; TCF/LEF, T-cell factor/lymphoid enhancer-binding factor; TNFα, tumor necrosis factor alpha; TX, thromboxane; VEGF, vascular endothelial growth factor.

**Figure 2 cancers-17-00520-f002:**
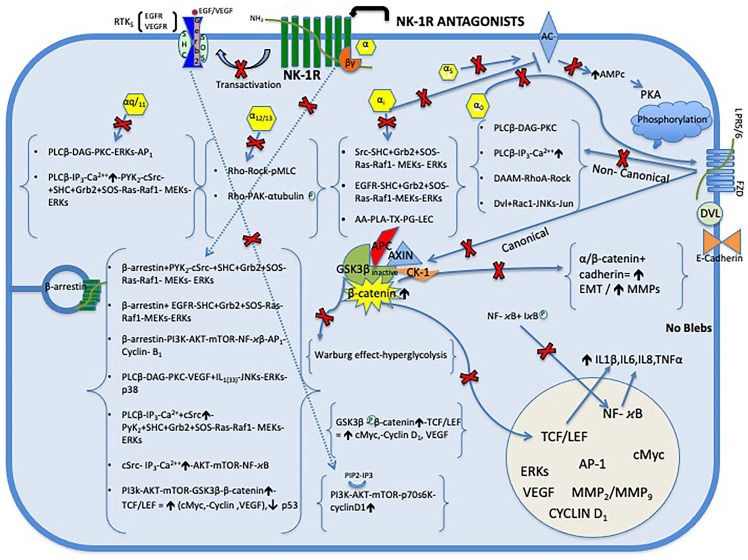
NK-1R antagonists, via the NK-1R, block these pathways and inhibit the effects mediated by SP on tumors. Abbreviations: AA, arachidonic acid; AC, adenylate cyclase; AKT or PKB, protein kinase B; APC, adenomatous polyposis coli; CK-1, casein kinase 1; DAAM, disheveled-associated activator of morphogenesis; DAG, diacylglycerol; Dvl; disheveled; EGF, epidermal growth factor; EGFR, epidermal growth factor receptor; EMT, epithelial–mesenchymal transition; ERK1/2, extracellular signal-regulated protein kinase; Fzd, frizzled; GSK3β, glycogen synthase kinase 3β; HK-1, hemokinin-1; IP3, inositol trisphosphate; LEC, leukotriene; LRP5/6, low-density lipoprotein receptor-related protein 5/6; MAPKs, mitogen-activated protein kinases; MEKs, mitogen-activated protein kinase kinases; MMPs, matrix metalloproteinases; NF-κB, nuclear factor kappa B; PKC, protein kinase C; PG, prostaglandin; pMLC, myosin light-chain kinase; PI3K, phosphoinositide 3-kinase; PLA, phospholipase A; PLC β, 1-phosphatidylinositol-4,5-bisphosphate phospholipase beta; ROCK, Rho-associated kinase; RTK, receptor tyrosine kinase; TCF/LEF, T-cell factor/lymphoid enhancer-binding factor; TNFα, tumor necrosis factor alpha; TX, thromboxane; VEGF, vascular endothelial growth factor.

**Table 1 cancers-17-00520-t001:** Autocrine, paracrine, intracrine, and endocrine function of SP in glioma.

SP Functions	SP Mechanisms	SP Actions
Autocrine	SP is synthesized in the cytoplasm of glioma cells and released into the extracellular space and binds to the NK-1R of the same glioma cell [10].	Mitogenesis and antiapoptotic effect in glioma cells [8].
Paracrine	SP is synthesized in the cytoplasm of glioma cells and released into the extracellular space and binds to NK-1Rs on other gliomas and inflammatory and endothelial cells [10].	Mitogenesis and antiapoptotic effect in gliomas, inflammatory and endothelial cells. Release inflammatory cytokines [8,32,39].
Intracrine	SP is synthesized in the cytoplasm of glioma cells and subsequently migrates to the nucleus, performing a nuclear function in glioma cells [10].	Transcription factor and mitogenesis [38,41].
Endocrine	The glioma tumor mass synthesizes and releases SP into the bloodstream, behaving like an endocrine organ and increasing plasma levels of SP [7,10].	Increased SP plasma levels [18].

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
