# Peer review of "Radiotherapy Plus the Neurokinin-1 Receptor Antagonist Aprepitant: A Potent Therapeutic Strategy for the Treatment of Diffuse Intrinsic Pontine Glioma"

_cancers, 2025, doi:10.3390/cancers17030520_

Round 1

Reviewer 1 Report

Comments and Suggestions for Authors

The review article submitted for review concerns the use of the neurokinin receptor-1 (NK-1R) system and its peptide and/or non-peptide agonist or antagonist ligands in cancer therapy, with particular attention paid to brain tumors. This topic has been the subject of research by many research groups for years, but new combined therapies are still sought to achieve greater treatment effectiveness and reduce harmful side effects.

Overall, the article fits perfectly into the subject of the Cancers journal, and the descriptions of signaling pathways (mechanisms) describing ligand-receptor interactions are a particularly valuable page.

However, in my opinion, there are shortcomings in the article that should be corrected.

A major shortcoming of the article is the insufficiently complete review of the cited literature. As I wrote, the subject of the application of the NK-1 receptor/ligand system in the diagnosis and treatment of cancers, especially brain tumors, has been the subject of research by many research groups, e.g. the group of Gniazdowska et al. The article sent for review (introduction, lines 56-73) concerns such research problems as the review article Pharmaceutics 2019, 11, 443, which should definitely be included in the cited literature.

Moreover, the Gniazdowska et al. research group has been dealing for years with the use of the non-peptide NK-1 receptor antagonist aprepitant in the design and application of a diagnostic/therapeutic radiopharmaceutical for the treatment of glioblastoma multiforme of the brain [Molecules 2020, 25, 3756; Int. J. Mol. Sci. 2022, 23, 1214; Pharmaceutics 2022, 14, 607]. These works concern the field of radiopharmacy (which the authors of the article submitted for review do not deal with), however, in my opinion, they should also be included in a review article.

Authors are also asked to check the text of the article carefully, e.g.:

·       Lines: 536 – 541; 598 - 599; 667 - 669; 598 - 599; 667-669 – the meaning and syntax of sentences;

·       Lines 391-392 and 396-397 – repeated sentence;

·       Lines 419, 530, 617 – cited literature

Author Response

All the changes in the new Ms appear in red.

 Reviewer 1

Response to Reviewer 1

We thank the Reviewer for the meticulous review and for the encouraging comment.

The review article submitted for review concerns the use of the neurokinin receptor-1 (NK-1R) system and its peptide and/or non-peptide agonist or antagonist ligands in cancer therapy, with particular attention paid to brain tumors. This topic has been the subject of research by many research groups for years, but new combined therapies are still sought to achieve greater treatment effectiveness and reduce harmful side effects.

Overall, the article fits perfectly into the subject of the Cancers journal, and the descriptions of signalling pathways (mechanisms) describing ligand-receptor interactions are a particularly valuable page.

However, in my opinion, there are shortcomings in the article that should be corrected.

A major shortcoming of the article is the insufficiently complete review of the cited literature. As I wrote, the subject of the application of the NK-1 receptor/ligand system in the diagnosis and treatment of cancers, especially brain tumors, has been the subject of research by many research groups, e.g. the group of Gniazdowska et al. The article sent for review (introduction, lines 56-73) concerns such research problems as the review article Pharmaceutics 2019, 11, 443, which should definitely be included in the cited literature.

We apologize for that oversight; it has now been included (see lines 70-77 in the in the new Ms)

Moreover, the Gniazdowska et al. research group has been dealing for years with the use of the non-peptide NK-1 receptor antagonist aprepitant in the design and application of a diagnostic/therapeutic radiopharmaceutical for the treatment of glioblastoma multiforme of the brain [Molecules 2020, 25, 3756; Int. J. Mol. Sci. 2022, 23, 1214; Pharmaceutics 2022, 14, 607]. These works concern the field of radiopharmacy (which the authors of the article submitted for review do not deal with), however, in my opinion, they should also be included in a review article.

The above papers have been included in the text in a new section called 10.8. SP/NK1R system in targeted tumor therapy with radionuclides in DIPG. See new section (see lines 711-733 in the new Ms).

Authors are also asked to check the text of the article carefully, e.g.:

  • Lines: 536 – 541; 598 - 599; 667-669 – the meaning and syntax of sentences. It has been done (see lines 546-550; 609-611; 689-691 in the new Ms).
  • Lines 391-392 and 396-397 – repeated sentence.One sentence has been removed (see lines 399-404 in new Ms)
  • Lines 419, 530 [J Bioscience and Glioma Muñoz], 617 [Niu]. – cited literature. It has been corrected (see line 426, 538, 627 in the new Ms).

Reviewer 2 Report

Comments and Suggestions for Authors

An interesting manuscript, but it seems to me that this manuscript has two conceptual problems.
First. The point is that the idea of ​​using neurokinin-1 receptor antagonists has been in the air for quite a long time. What exactly happened recently? Why, after 20 years (or so) of discussing the problem, did the authors decide to write a review? What new things did the authors bring to the discussion? What other points of view are there on the problem? What potential developments does the authors' idea compete with? How is it different for the better? Or maybe for the worse? I have a strong impression that the manuscript is written in such a way that the authors formulated the idea for the first time, and before them there was a vacuum - this is not true. It seems to me that the authors need to find the strength to correct the text.
Second. Of the entire huge manuscript, only subsection 10.7 is devoted to radiotherapy... Radiosensitizers, radioprotectors that do not accumulate in the tumor, or symptomatic therapy are usually tried with radiotherapy. It is obvious that the idea presented by the authors does not fit into any of the strategies described above. This means only one thing, that the word radiotherapy in the title and in the text is superfluous. Neurokinin-1 receptor antagonists can be an adjuvant therapy or something else, but they are definitely not related to tumor radiotherapy. I understand that the authors tried to introduce something new, but in this case it did not work. My doubts are also related to the high degree of borrowing of the text and not always up-to-date references to the literature.
I also have suggestions for improving the manuscript, I hope my ideas will help the authors.

1. The authors describe the functions of NK-1R receptors in quite a lot of detail. However, the manuscript contains almost no data on NK-2R and NK-3R receptors. I think this needs to be corrected. Please, post at least brief information on NK-2R and NK-3R receptors. Characterize the structural features, effectors, etc.

2. Chapter "3. SP is involved in many types of signaling in glioma" is written quite clearly, but the chapter relies on several literary references, some of which are more than a third of a century old. Moreover, the literary references are concentrated in several and it is not at all clear which of the references supports this or that thesis of the authors. It seems to me that the situation needs to be corrected. Moreover, I believe that this chapter is key to understanding the whole idea. Perhaps the authors need to make a table in this chapter, that would be great!

3. The authors believe that the use of cytokine and chemokine blockers, as well as non-steroidal anti-inflammatory drugs can be useful in the chemoprevention of malignant diseases. Perhaps the authors need to explain and significantly concretize their idea.

4. The drugs aprepitant, fosaprepitant, netupitant, fosnetupitant, rolapitant, tradipitant are non-peptide NK-1R antagonists. According to the authors' idea, these compounds should have a significant effect on the development of the disease. Is this true? If yes, then write how each drug has an effect. If no, then you need to write an explanation why it has no effect in the authors' opinion. This is a rather important question and cannot be ignored. I would also suggest making a mini table.

Minor points
Figure 1 should be located after the first mention on page 3, not on page 6. This is not convenient for the reader.
Potassium ions are monovalent, please correct on line 108 (K++)
Please duplicate all abbreviations under Figure 2, so that readers do not have to flip through the manuscript and look for abbreviations each time.
On line 249, the word Indeed should not be written in bold.
Chapter titles cannot contain references to figures. Correct.
Remove italics from lines 537-540.
Line 530, instead of a reference it says "J Bioscience and Glioma Muñoz" - needs to be corrected.

Author Response

All the changes in the new Ms appear in red.

Reviewer 2

Response to Reviewer 2

 We thank the reviewer for his meticulous review and encouraging comments.

An interesting manuscript, but it seems to me that this manuscript has two conceptual problems.
First. The point is that the idea of using neurokinin-1 receptor antagonists has been in the air for quite a long time. What exactly happened recently? Why, after 20 years (or so) of discussing the problem, did the authors decide to write a review?

For these reasons among others, we have decided to write this review: it has been reported that in a patient with lung cancer and COPD, not suitable to surgical treatment or chemotherapy, a combined treatment with palliative radiotherapy and compassionate use of aprepitant (1140 mg/day for 45 days) was used, the tumor mass disappeared six months after treatment and no serious side effects were observed [Mol. Clin. Oncol. 2019, 11]. In addition, it has recently been shown that NK-1R is essential for glioma cell viability and is not essential in normal non-tumor cells. However, SP is not essential for the viability of glioma cells. Furthermore, both SP and the NK-1R were localized in the nucleus and cytoplasm of glioma cells [Biomed. Res. Int. 2022, 6291504].

What new things did the authors bring to the discussion? What other points of view are there on the problem? What potential developments does the authors' idea compete with? How is it different for the better? Or maybe for the worse? I have a strong impression that the manuscript is written in such a way that the authors formulated the idea for the first time, and before them there was a vacuum - this is not true. It seems to me that the authors need to find the strength to correct the text.

The strength of this review is that, compared to other reviews, it provides a comprehensive view linking the NK-1R and the molecular signaling that controls glioma cell proliferation, migration, and invasion and drives angiogenesis and inflammation in the tumor microenvironment. The article discusses in depth the clinical results of clinical trials using radiotherapy in combination with NK-1R antagonists in different types of cancer and various pathological conditions. The aim of the article is to summarize the available data and justify the need and potential clinical efficacy in glioma (DIPG) patients of a combination of radiotherapy and the NK-1R antagonist drug aprepitant or other similar drugs with non-toxic and promising antitumor activity profiles. Compared with other reviews on gliomas and NK-1R antagonists written by one of the authors, the current manuscript offers a novel concept that consistently justifies the combination of radiotherapy plus aprepitant, a clinically well-studied and safe NK-1R antagonist, in children with life-threatening DIPG.

Second. Of the entire huge manuscript, only subsection 10.7 is devoted to radiotherapy... Radiosensitizers, radioprotectors that do not accumulate in the tumor, or symptomatic therapy are usually tried with radiotherapy. It is obvious that the idea presented by the authors does not fit into any of the strategies described above. This means only one thing, that the word radiotherapy in the title and in the text is superfluous. Neurokinin-1 receptor antagonists can be an adjuvant therapy or something else, but they are definitely not related to tumor radiotherapy. I understand that the authors tried to introduce something new, but in this case it did not work. My doubts are also related to the high degree of borrowing of the text and not always up-to-date references to the literature.

In accordance with the reviewer's suggestion, we have increased the amount of text on radiotherapy with current references. It is well known that radiotherapy is the standard treatment for DIPG; however, it is only palliative. radiotherapy is expected to increase patient survival by approximately 3 months on average. DIPG remains universally fatal (lines 45-48 in the new Ms). Thus, it is necessary to combine it with other drugs that have a direct antitumor effect on glioma, that diffuse the blood-brain barrier, that are safe, that produce radiosensitization, as well as that reduce the side effects of radiotherapy and that are anti-inflammatory for both tumor inflammation and that produced by radiotherapy (see lines 674-710 in the new Ms). Furthermore, for a more comprehensive review, now we have included a new section 10.8. SP/NK-1R system in targeted tumor therapy with radionuclides in DIPG. This section summary the possibility of a new treatment of the SP/NK1R system in targeted tumor therapy with radionuclides in DIPG (see lines 711-733 in the new Ms).

I also have suggestions for improving the manuscript, I hope my ideas will help the authors.

Thank you very much for your thorough and thoughtful suggestions to improve the review.

  1. The authors describe the functions of NK-1R receptors in quite a lot of detail. However, the manuscript contains almost no data on NK-2R and NK-3R receptors. I think this needs to be corrected. Please, post at least brief information on NK-2R and NK-3R receptors. Characterize the structural features, effectors, etc.

We have included brief information on this subject (see lines 81-85 in the new Ms).

  1. Chapter "3. SP is involved in many types of signaling in glioma" is written quite clearly, but the chapter relies on several literary references, some of which are more than a third of a century old. Moreover, the literary references are concentrated in several and it is not at all clear which of the references supports this or that thesis of the authors. It seems to me that the situation needs to be corrected. Moreover, I believe that this chapter is key to understanding the whole idea. Perhaps the authors need to make a table in this chapter, that would be great!

The reason for having references that are more than a third of a century old is that it is necessary to know the sequence of SP (it contains two phenylalanine residues) [Nat. New. Biol. 1971, 232] in order to understand the possible binding to the TATA box through those two phenylalanine residues in a similar way as the TATA binding protein does (see modified paragraph, lines 190-193 in the new Ms). Now, for a better understanding of the chapter we have made a table as suggested by the reviewer (see Table 1 in the new Ms).

  1. The authors believe that the use of cytokine and chemokine blockers, as well as non-steroidal anti-inflammatory drugs can be useful in the chemoprevention of malignant diseases. Perhaps the authors need to explain and significantly concretize their idea.

It has been reported the idea of using cytokine and chemokine blockers [Balkwill and Mantovani Lancet. 2001]. We agree with this idea and propose the use of NK-1R antagonists as anti-inflammatory drugs to reduce the concentrations of cytokines and chemokines. See lines 319-352 and 663-665 in the new Ms.

  1. The drugs aprepitant, fosaprepitant, netupitant, fosnetupitant, rolapitant, tradipitant are non-peptide NK-1R antagonists. According to the authors' idea, these compounds should have a significant effect on the development of the disease. Is this true? If yes, then write how each drug has an effect. If no, then you need to write an explanation why it has no effect in the authors' opinion. This is a rather important question and cannot be ignored. I would also suggest making a mini table.

Non-peptide NK-1R antagonists include compounds that exhibit similar stereochemical characteristics and different chemical compositions; Therapeutic activity is linked to both the affinity of these compounds for the NK-1R and the concentration of the compounds. (see lines 536-538 in the new Ms). NK-1R antagonists CP-96,345, L-733,060, L-732,138, MEN-11,467, MEN-11,149, the drugs aprepitant, and cyclosporine A (CsA) have been shown to have antitumor activity in glioma. All of them produce inhibition of proliferation and death by apoptosis in glioma cells. In addition, it is known that inducing apoptosis in tumor cells is a key anti-cancer treatment strategy (see lines 552-556 in the new Ms). Unfortunately, the NK-1R antagonist drugs netupitant, fosnetupitant, rolapitant and tradipitant are currently only known for their anti-nausea and vomiting activities and their antitumor activity in gliomas has not been studied to date (see lines 541-543 in the new Ms), so it is not possible to make a table on this topic as has been suggested.

Minor points

Figure 1 should be located after the first mention on page 3, not on page 6. This is not convenient for the reader.

It has been done.

Potassium ions are monovalent, please correct on line 108 (K++).

It has been corrected (see line 115 in the new Ms).

Please duplicate all abbreviations under Figure 2, so that readers do not have to flip through the manuscript and look for abbreviations each time.

It has been done.

On line 249, the word Indeed should not be written in bold.

It has been corrected (see line 274 in the new Ms).

Chapter titles cannot contain references to figures. Correct.

It has been corrected.

Remove italics from lines 537-540.

It has been changed (see lines 546-550 in the new Ms).

Line 530, instead of a reference it says "J Bioscience and Glioma Muñoz" - needs to be corrected.

It has been corrected (see line 538 in the new Ms).

Reviewer 3 Report

Comments and Suggestions for Authors

Dramatically decreased of a lifespan of kids with a brain-stem pontine glioma that lasts only 16-24 mo. under any treatment and 90-percent recurrence rate of post-treatment makes significant any promising efforts to improve the current treatment of glioma. The main merit of the review is that it, compared to other reviews, provides comprehensive view linking neurokinin-1 receptor (NK-1R) and molecular signaling that control proliferation, migration, invasion of glioma cells and drives angiogenesis and inflammation in tumor microenvironment. The paper minutely discusses clinical results of clinical trials that use radiotherapy (RT) in combination with NK-1R antagonist in different cancers and various pathological conditions. The paper logically substantiates role of substance P as a potent ligand for NK-1R in various signaling and metabolism in glioma and clearly demonstrate the existence of multiple NK-1R-dependent pathways significant for glioma growth, proliferation and migration/invasion/metastases, as well as for angiogenesis and inflammation. The main feature of the paper is to summarize available data and justify the need and potential clinical efficiency in glioma patients of a combination of RT and NK-1R antagonist antidepressant aprepitant or other similarly repurposed drugs with a similar non-toxic and promising anti-cancer activity profiles. Compared to other several reviewes on glioma and NK1-R antagonists written by one of the authors  the current manuscript provides a new concept that consistently   justifies combination of RT and well-studied clinically and safe NK1-R antagonist, aprepitant, in children with life threatening diffused pontine glioma.

Author Response

Reviewer 3

We thank the Reviewer for the meticulous review and for the encouraging comment.  Dramatically decreased of a lifespan of kids with a brain-stem pontine glioma that lasts only 16-24 mo. under any treatment and 90-percent recurrence rate of post-treatment makes significant any promising efforts to improve the current treatment of glioma. The main merit of the review is that it, compared to other reviews, provides comprehensive view linking neurokinin-1 receptor (NK-1R) and molecular signaling that control proliferation, migration, invasion of glioma cells and drives angiogenesis and inflammation in tumor microenvironment. The paper minutely discusses clinical results of clinical trials that use radiotherapy (RT) in combination with NK-1R antagonist in different cancers and various pathological conditions. The paper logically substantiates role of substance P as a potent ligand for NK-1R in various signaling and metabolism in glioma and clearly demonstrate the existence of multiple NK-1R-dependent pathways significant for glioma growth, proliferation and migration/invasion/metastases, as well as for angiogenesis and inflammation. The main feature of the paper is to summarize available data and justify the need and potential clinical efficiency in glioma patients of a combination of RT and NK-1R antagonist antidepressant aprepitant or other similarly repurposed drugs with a similar non-toxic and promising anti-cancer activity profiles. Compared to other several reviews on glioma and NK1-R antagonists written by one of the authors the current manuscript provides a new concept that consistently   justifies combination of RT and well-studied clinically and safe NK1-R antagonist, aprepitant, in children with life threatening diffused pontine glioma.

The authors deeply appreciate your words and encourage us to continue working on this topic.

Round 2

Reviewer 2 Report

Comments and Suggestions for Authors

The manuscript has been significantly improved. I recommend it for publication.